

# Comparative proteomics of *Rhizopus delemar* ATCC 20344 unravels the role of amino acid catabolism in fumarate accumulation

Dorett I. Odoni[1,*], Juan A. Tamayo-Ramos[1,*], Jasper Sloothaak[1], Ruben G.A. van Heck[1], Vitor A.P. Martins dos Santos[1,2], Leo H. de Graaff[1,†], Maria Suarez-Diez[1] and Peter J. Schaap[1]

[1] Laboratory of Systems and Synthetic Biology, Wageningen University & Research, Wageningen, The Netherlands
[2] LifeGlimmer GmBH, Berlin, Germany
[*] These authors contributed equally to this work.
[†] Deceased.

## ABSTRACT

The filamentous fungus *Rhizopus delemar* naturally accumulates relatively high amounts of fumarate. Although the culture conditions that increase fumarate yields are well established, the network underlying the accumulation of fumarate is not yet fully understood. We set out to increase the knowledge about fumarate accumulation in *R. delemar*. To this end, we combined a transcriptomics and proteomics approach to identify key metabolic pathways involved in fumarate production in *R. delemar*, and propose that a substantial part of the fumarate accumulated in *R. delemar* during nitrogen starvation results from the urea cycle due to amino acid catabolism.

## INTRODUCTION

Fumarate, a dicarboxylic acid, is an important building block chemical for a number of high-value chemicals and materials. Amongst the microorganisms identified to be natural fumarate producers, the filamentous fungus *Rhizopus delemar* has the highest product yields (*Foster & Waksman, 1939*). The most important factor influencing fumarate production in *R. delemar* is a high carbon:nitrogen ratio; extracellular fumarate accumulation happens after the growth phase, and especially when the nitrogen in the medium has been depleted (*Magnuson & Lasure, 2004*; *Goldberg, Rokem & Pines, 2006*). The choice of nitrogen source has been reported to influence the final fumarate yield (*Foster & Waksman, 1939*; *Rhodes et al., 1959*; *Zhang & Jin JMK, 2007*), but so far no consensus on these influences has been reached. Another important factor influencing fumarate production in *R. delemar* is oxygen availability (*Foster & Waksman, 1939*; *Foster et al., 1949*). Under fumarate producing conditions, *R. delemar* forms ethanol and other undesired fermentation by-products (*Wright, Longacre & Reimers, 1996*), directing carbon

Corresponding author
Peter J. Schaap, peter.schaap@wur.nl

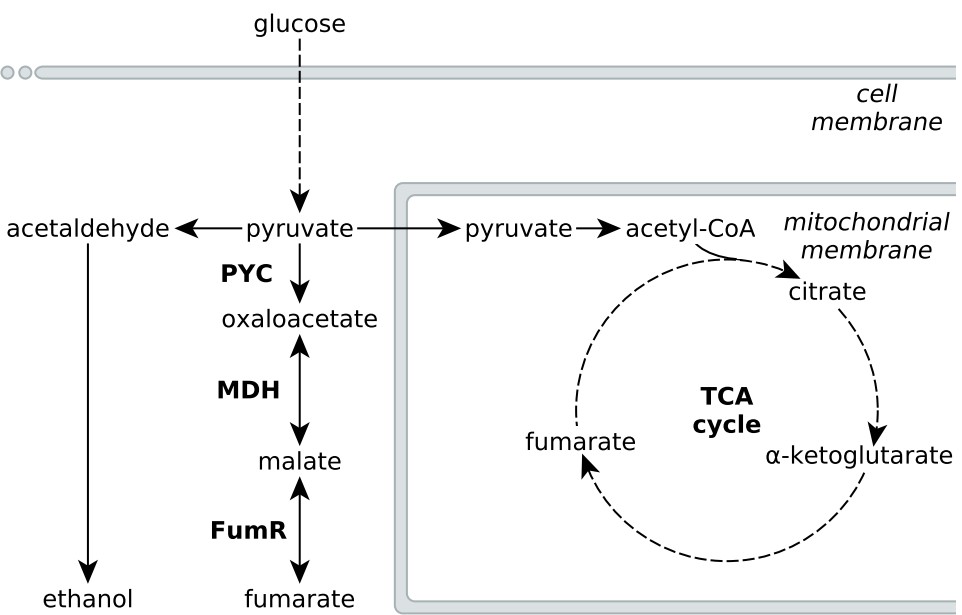

**Figure 1 Metabolic pathways involved in fumarate metabolism in *R. delemar*.** Metabolic flux of *R. delemar* is predominantly directed towards fumarate (under aerobic conditions) or ethanol (under anaerobic conditions). The enzymes of the reductive TCA cycle are indicated in the scheme: PYC, pyruvate carboxylase; MDH, L-malate dehydrogenase and FumR, fumarase.

away from fumarate (Fig. 1). Higher oxygen levels limit the amount of ethanol produced, and thus lead to higher fumarate yields. Fumarate production by fermentation has been extensively reviewed (*Straathof & van Gulik, 2012*; *Roa Engel et al., 2008*).

Although the culture conditions that increase fumarate accumulation in *R. delemar* are well established, natural product titers still cannot compete with chemical fumarate synthesis. To increase the amount of fumarate produced, *R. delemar* has been genetically modified (*Meussen et al., 2012*; *Zhang, Skory & Yang, 2012*; *Zhang & Yang, 2012*), but the occurrence of an ancestral whole-genome duplication as well as more recent gene-duplication events complicate the genetic engineering of *R. delemar* (*Ibrahim et al., 2009*). A more promising approach for biological fumarate production would thus be rewiring the metabolism of a genetically more amenable cell-factory, based on *R. delemar* fumarate synthesis pathways.

Metabolic engineering approaches to increase fumarate production in microbial cell-factories that do not naturally accumulate high amounts of fumarate would greatly benefit from an in-depth understanding of the underlying metabolic pathways that affect the accumulation of fumarate in the natural fumarate producer *R. delemar*, as well as possible causes for this accumulation. Fumarate can be found as an intermediate in various different metabolic subsystems, and a number of pathways have been investigated for fumarate production in several microbial cell-factories such as *Saccharomyces cerevisiae*, *Torulopsis glabrata*, *Scheffersomyces stipitis* and *Escherichia coli* (*Chen et al., 2015*; *Li et al., 2014*; *Song et al., 2013*; *Wei et al., 2015*; *Xu et al., 2013*; *Xu, Liu & Chen, 2012*; *Xu et al., 2012*; *Zhang et al., 2015*; *Chen, Zhu & Liu, 2016*). Despite the large number of pathways

leading to fumarate, the current consensus is that the reductive route of the TCA cycle in the cytosol is responsible for fumarate accumulation in *R. delemar* (*Romano, Bright & Scott, 1967*; *Overman & Roman, 1969*; *Osmani & Scrutton, 1985*; *Kenealy et al., 1986*; *Peleg et al., 1989*). The reductive TCA cycle comprises pyruvate carboxylase (PYC), L-malate dehydrogenase (MDH) and fumarase (FumR), in which pyruvate is consecutively converted to oxaloacetate, L-malate and fumarate (Fig. 1).

A controversial aspect of this pathway model is FumR. While overexpression of *pyc* and *mdh* gave the expected increase of fumarate in *R. delemar* and *Saccharomyces cerevisiae* (*Zhang, Skory & Yang, 2012*; *Xu et al., 2012*), overexpression of *fumR* in *R. delemar* as well as the introduction of *fumR* in *S. cerevisiae* and *A. niger* was reported to result in more L-malate rather than the accumulation of fumarate (*Zhang & Yang, 2012*; *Xu et al., 2012*; *De Jongh & Nielsen, 2008*). There has been debate about the role of FumR in fumarate accumulation, discussed by *Meussen et al. (2012)*. In summary, the reaction kinetics of FumR favour the conversion of fumarate to L-malate rather than the reverse direction, and FumR activity of acid-producing mycelium is completely blocked in the presence of 2 mM fumarate (note that the data on FumR inhibition have never been published, but that this finding is cited by *Goldberg, Rokem & Pines, 2006*; *Meussen et al., 2012*). This suggests the presence of an alternative pathway that is responsible for fumarate accumulation in *R. delemar*.

In this study, we aim to provide a holistic overview of the pathways involved in fumarate accumulation in the natural fumarate producer *R. delemar*. To this end, we cultured the *R. delemar* strain ATCC 20344 under nitrogen starvation conditions, and varied oxygen availability to induce high (aerobic) and low (anaerobic) fumarate production. Combining transcriptomic and proteomic data obtained from the two conditions, we revealed the relationship between nitrogen metabolism and fumarate accumulation in *R. delemar*, mediated by the urea cycle.

## MATERIALS AND METHODS

### Strains, media and culture conditions

We selected the *R. delemar* strain ATCC 20344 (a kind gift from Adrie J.J. Straathof, Delft University of Technology) to study fumarate production. Note that *R. delemar* is more commonly known as *R. oryzae* (also *Rhizopus nigricans* and *Rhizopus arrhizus*) (*Abe et al., 2007*). Depending on the organic acid produced when grown on D-glucose, it is divided into two phylogenetically distinct types: type I strains, which produce primarily L-lactate, and type II strains, which produce mainly fumarate and L-malate (*Abe et al., 2003*). Complete medium agar plates containing 0.3% (w/v) yeast extract, 0.3% (w/v) malt extract, 0.3% (w/v) peptone, 2% (w/v) glycerol, and 2% (w/v) agar were used to generate spores. Mycelial biomass was produced using pre-culture medium containing 1% (w/v) D-glucose, 0.21% (w/v) urea, 0.06% (w/v) $KH_2PO$, 0.05% (w/v) $MgSO_4.7H_2O$ and 0.0018% (w/v) $ZnSO_4.7H_2O$. Approximately $10^6$ spores/mL were inoculated in 1L Erlenmeyer flasks containing 500 mL of pre-culture medium and cultivations were carried out at 35 °C and 250 rpm for 24 h. The mycelium obtained was washed with demineralized water and

transferred ($\approx$25 g of wet biomass) to production medium, which contained 10% (w/v) D-glucose, 0.06% (w/v) $KH_2PO_4$, 0.05% (w/v) $MgSO_4.7H_2O$, 0.0018% (w/v) $ZnSO_4.7H_2O$, and 1% (w/v) $CaCO_3$ (used as a neutralizing agent). Batch fermentations were performed at 35 °C and 600 rpm in 2.5 L fermentors (Applikon, Schiedam, The Netherlands), with a working volume of 1.75 L. Antifoam 204 was added to each fermentor (85 µL). The fermentation medium was aerated with 1.0 L/L min, either with filtered air or $N_2$ gas.

## Metabolite analysis using HPLC

Extracellular metabolite concentrations were determined by high-performance liquid chromatography (HPLC). A Thermo Accela HPLC, equipped with a Shodex KC-811 column, and coupled to a refractive index detector (Spectrasystem RI-150, sample frequency 5.00032 Hz) and a UV–VIS detector (Spectrasytem UV1000, $\lambda = 210$ nm), was used. Separations were performed by isocratic elution with 0.01 N $H_2SO_4$ at a flow rate of 0.8 mL/min. Crotonic acid (6 mM) was used as an internal standard.

## RNA isolation and quality control

Frozen mycelium ($\approx$100 mg) of *R. delemar* ATCC 20344 was submerged in 1 mL of Trizol reagent in a 2 mL tube, prefilled with a mix of glass beads with the following diameters: 1 mm (0.25 g), 0.1 mm (0.37 g), 5 mm (1 bead). Mycelium samples were disrupted using a FastPrep-24 Instrument (MP). After disruption, 200 µL of chloroform were added and the mix was homogenated for 10 s. The mix was poured into phase-lock gel tubes (2 mL), and centrifuged at maximum speed in a table-top centrifuge. The RNA present in the water phase was purified using the RNeasy Mini Kit (Qiagen), following the manufacturer's instructions.

RNA integrity was assessed with an Experion system (Bio-Rad), and only high quality samples (RIN value $\geq$8) were selected for whole transcriptome shotgun sequencing.

## RNA sequencing and quality check

Illumina RNA sequencing (RNA seq) using the Casava pipeline version 1.8.2 and subsequent quality analysis of the FASTQ sequence reads was performed by BaseClear (Leiden, The Netherlands). The number of reads obtained was 20′539′199 for the aerobic and 24′519′028 for the anaerobic condition, with an average quality score (Phred) of 37.59 and 37.91, respectively. The raw data has been submitted to the European Nucleotide Archive (ENA), and can be found under the accession number PRJEB14210 (http://www.ebi.ac.uk/ena/data/view/PRJEB14210).

## RNA seq data processing

Following the wokflow suggested and validated by *Davids et al. (2016)*, the RNA seq reads were filtered using SortMeRNA v1.9 (*Kopylova, Noé & Touzet, 2012*), cutadapt v1.2.1 (*Martin, 2011*) and PRINSEQ v0.20.2 (*Schmieder & Edwards, 2011*). *De novo* assembly of the reads that passed the quality filtering was performed using the IDBA-UD assembler v1.1 (*Peng et al., 2012*). Read mapping and transcript coverage calculations were performed using Bowtie2 v 2.2.2 (*Langmead & Salzberg, 2012*) and BEDTools (*Quinlan & Hall, 2010*). Note that in contrast to the proteomics analysis, which was performed on both biological

replicates, only one biological replicate per condition was sent for RNA sequencing. The average nucleotide coverage is thus an indication of the transcript levels of a given transcript the time of sampling, not the average of two biological replicates. A more extensive description of the RNA seq data processing, including example commands for every tool used, is given in File S1.

## Preparation of cell free extracts for proteomic analysis

*R. delemar* ATCC 20344 mycelium samples (2–3 g, press-dried), were washed with an ice-cold 20 mM HEPES buffer pH 7.6, containing 150 mM NaCl, and resuspended in the same solution containing 1% (v/v) protease inhibitor cocktail for yeast and fungi (Sigma). Mycelium suspensions were immediately disrupted using a French press (8,000 psi). Cell free extracts were centrifuged for 5 min at low speed (500 g), in order to remove unbroken cells and pellet debris. The remaining supernatants were further processed for LC-MS/MS analysis.

## Sample preparation for LC-MS/MS

The protein content of the *R. delemar* ATCC 20344 cell free extracts was determined using the BCA protein assay (Thermo Fisher). Membrane-bound proteins were solubilised by mixing volumes of each sample, containing 25 $\mu$g of protein, with equal volumes of a 2×solution of 20 mM HEPES pH 7.6, containing 1 M 6-aminocaproic acid and 10 g $L^{-1}$ of n-dodecyl-beta-D-maltoside. Cell free extract-detergent mixes were incubated in a thermoblock for 1 h at 20 °C and vigorous stirring (1,000 rpm). Afterwards, samples were sonicated in a water bath for 15 min, and finally they were centrifuged at 22,000 g, in a benchtop centrifuge, for 30 min. Obtained supernatants were subsequently concentrated using Microcon YM-10 columns (cutoff, 10 kDa; Millipore, Eschborn, Germany).

Samples from each biological replicate and culture condition were loaded into a 12% SDS-polyacrylamide gel, which was run until the loaded samples entered the gel. The gel was stained according to the manufacturer's instructions using Page Blue staining (Fermentas) and rinsed with ultrapure water. Each sample-gel lane was cut into one slice (approx. 1 cm$^2$), carefully sliced into smaller pieces of about 1 mm$^3$ and transferred into microcentrifuge tubes. Samples were destained and equilibrated through three washing steps using the following solutions: 50 mM ammonium bicarbonate (ABC) (incubated 5 min), ABC/acetonitrile (1:1, v/v) (incubated 5 min) and neat acetonitrile (incubated 5 min). These washing steps were repeated two times. The gel samples were then swelled in 10 mM dithiothreitol (DTT) for 20 min at 56 °C to reduce protein disulfide bonds. Subsequently, the DTT solutions were removed and samples were alkylated with 50 mM 2-chloroacetamide in ABC, for 20 min, at room temperature, in the dark. The 2-chloroacetamide solutions were removed, and samples were again washed twice with: neat acetonitrile (incubated 5 min), ABC (incubated 5 min) and neat acetonitrile (incubated 5 min). Approximately 150 $\mu$L of digestion buffer, containing sequencing grade modified trypsin (12.5 ng/$\mu$L) (Promega) in ABC, was added to each sample, making sure that all gel pieces were kept wet during digestion (adding, if necessary, additional ABC solution). Protein samples were digested overnight at 37 °C. Peptide digestion products were extracted

by adding 50 µL of 2% trifluoroacetic acid (TFA), followed by an incubation step in a thermoblock for 20 min, at room temperature and vigorous stirring (1,400 rpm). Gel pieces were then subjected to 20 s sonication in a water bath, centrifuged and supernatants were transferred to new tubes. The peptide extraction step was then repeated once by washing the gel pieces with buffer B (80% acetonitrile, 0.1% formic acid) followed by the mentioned incubation and sonication steps. Supernatants from both extractions were pooled and samples were placed in a vacuum centrifuge for acetonitrile evaporation util 20–40 µL were left. Finally, samples were acidified by addition of TFA (1:1, v/v) and peptide clean-up procedure, prior to LC-MS/MS analysis, was performed using the "STop And Go Extraction" procedure as described before (*Rappsilber, Ishihama & Mann, 2003*).

## Mass spectrometric measurements and proteomic data analysis

LC-MS/MS analysis was performed at the Radboud Proteomics Centre as described previously (*Rajala et al., 2015*). Measurements were performed by nanoflow reversed-phase C18 liquid chromatography (EASY nLC, Thermo Scientific) coupled online to a 7 Tesla linear ion trap Fourier-Transform ion cyclotron resonance mass spectrometer (LTQ FT Ultra, Thermo Scientific). The LC-MS/MS spectra obtained were identified and quantified using the maxQuant software (*Cox & Mann, 2008*). The peptides were mapped against the *in silico* proteomes of *R. delemar* ATCC 20344 (obtained from the transcriptomics experiment) and RA 99-880 (obtained from Genbank, Project ID: 13066 (*Ibrahim et al., 2009*)) with the default settings, described in (*Sloothaak et al., 2015*). Only proteins with 2 or more unique peptide hits were considered for further analysis. The mass spectrometry proteomics data have been deposited to the ProteomeXchange Consortium via the PRIDE (*Vizcaíno et al., 2016*) partner repository with the dataset identifier PXD004600.

## Metabolic pathway enrichment analysis

Metabolic enzymes were annotated using PRIAM (*Claudel-Renard et al., 2003*), and subsequently assigned to KEGG (*Kanehisa & Goto, 2000*; *Kanehisa et al., 2015*) pathways (see File S1 for details on the KEGG pathway mapping). Enrichment analysis of differentially expressed pathways was performed using the hypergeometric test implementation ("phyper") of the R software environment (*R Core T R, 2014*). We used the identified proteins that could be mapped to a KEGG pathway as the universe (with size $N = 277$). Note that the terms "differentially expressed" and "overexpressed" refer to differences in relative protein abundances, and denote a fold-change of 1.5 and 2 as lenient and stringent thresholds.

## RESULTS AND DISCUSSION

### Fumarate and ethanol production of ATCC 20344 grown under aerobic and anaerobic conditions

We chose to work with *R. delemar* ATCC 20344, henceforth referred to as ATCC 20344, for its ability to produce fumarate in high quantities (*Cao et al., 1996*). ATCC 20344 was grown in batch fermentations under nitrogen starved conditions. The fumarate production rate was controlled by either supplying filtered air to the culture medium (aerobic condition),

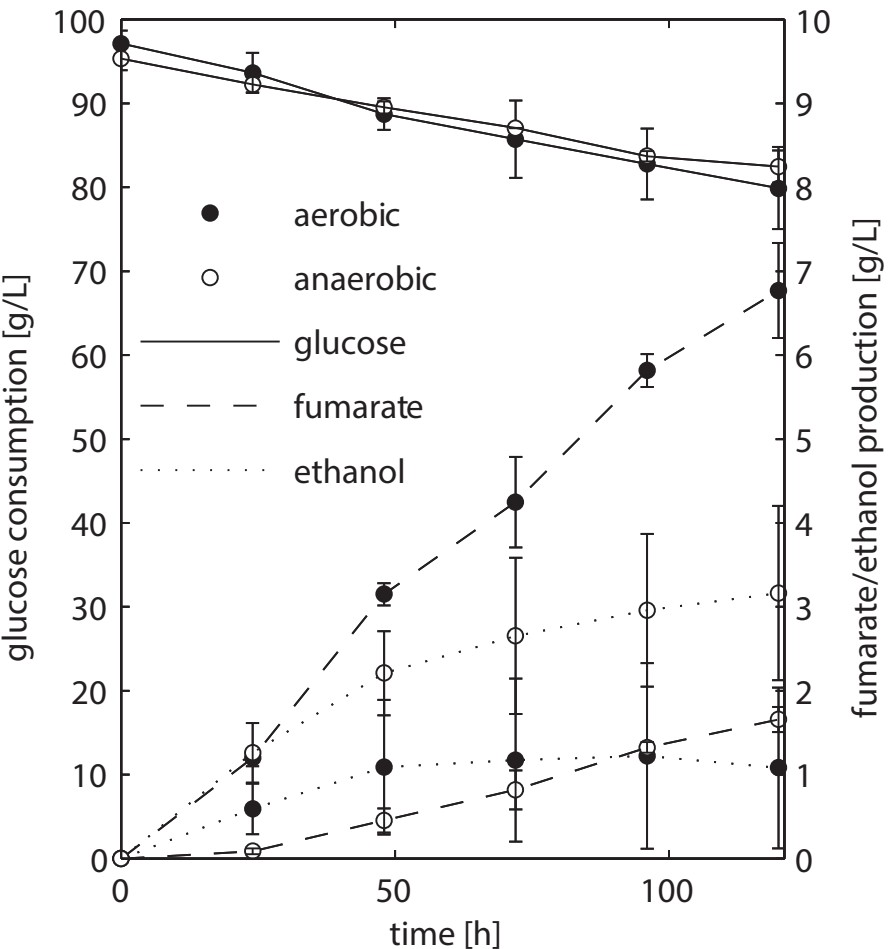

**Figure 2  HPLC analysis of fermentation broth of *R. delemar* ATCC 20344 grown under aerobic and anaerobic conditions.** Total D-glucose consumption and fumarate and ethanol production of *R. delemar* ATCC 20344. The measurement points show the average of two biological replicates.

or restricting the amount of oxygen by flushing the system with $N_2$ (anaerobic condition). D-glucose, fumarate and ethanol concentrations in the supernatant were measured *via* High Performance Liquid Chromatography (HPLC) to determine the time point with the largest difference in fumarate yield. The HPLC analysis showed comparable D-glucose consumption rates in the two conditions (Fig. 2), with an average of $0.15 \pm 0.03$ g/h and $0.11 \pm 0.03$ g/h for the aerobic and anaerobic condition, respectively. Fumarate production was higher in the aerobic condition, whereas in the anaerobic condition ethanol production prevailed. Note that *Lin & Wang (1991)* showed that *Rhizopus spp.* grow very poorly under absolute anaerobic conditions, but that most of the tested strains grew "quite well" under microaerobic conditions. The use of silicone tubing on our fermentors, which are slightly oxygen permeable even when flushed with pure nitrogen gas (*Weusthuis et al., 1994*), allowed ATCC 20344 to utilise glucose at the same rate as under aerobic conditions, while keeping fumarate production to a minimum. However, the amount of oxygen entering through the silicone tubing was below the detection limit of the probes measuring dissolved

**Table 1  Fumarate yields of ATCC 20344 grown under aerobic and anaerobic conditions.**

| Time [h] | Fumarate yield ± sd [g/g D-glucose] | |
| --- | --- | --- |
| | Aerobic | Anaerobic |
| 24 | 0.349 ± 0.055 | 0.029 ± 0.015 |
| 48 | 0.375 ± 0.001 | 0.077 ± 0.015 |
| 72 | 0.380 ± 0.055 | 0.098 ± 0.012 |
| 96 | 0.411 ± 0.063 | 0.115 ± 0.014 |
| 120 | 0.397 ± 0.044 | 0.132 ± 0.022 |

oxygen in the fermentors, and we refer to the two conditions as "aerobic" and "anaerobic" rather than "aerobic" and "microaerobic".

The fumarate yields (gram per gram substrate D-glucose consumed) are summarised in Table 1. The maximum fumarate yield (0.41 ± 0.06 g/g) in the aerobic condition was observed after 96 h of fermentation. A comparable yield (0.35 ± 0.05 g/g) was already observed after 24 h of fermentation. In contrast, the fumarate yield in the anaerobic condition increased continuously at a slow pace, being highest after 120 h of fermentation (0.13 ± 0.02 g/g). Thus, the largest difference in fumarate yield between the two conditions was observed at the start of the experiment, and we chose $t = 24$ h as the time point for the transcriptome and proteome analyses.

## Transcriptome and proteome of ATCC 20344 under high and low fumarate producing conditions

Enzyme activities, and thus metabolism, are affected by various factors such as post-translational modifications, allosteric control, and substrate availability. Metabolic fluxes can therefore not be inferred directly from protein abundances. Nevertheless, contrasting enzyme abundance levels between the high and low fumarate producing condition (ATCC 20344 snapshot proteomes) indicate differences in the metabolic state of ATCC 20344 at the time point of sampling. To determine differential protein abundances *via* LC-MS/MS, a reference proteome database is required for peptide mass fingerprinting. To date, *R. delemar* RA 99-880, henceforth referred to as RA 99-880, is the only fully sequenced *R. delemar* strain of which also the proteome is publicly available (*Ibrahim et al., 2009*). However, if the RA 99-880 reference proteome is used as only reference database, conservative amino acid substitutions in ATCC 20344 will reduce the sensitivity, as protein identification relies on an exact peptide mass. To provide a complete database of the metabolic potential as well as an overview of the metabolic state of ATCC 20344 under high and low fumarate producing conditions, we combined transcriptomic and proteomic data of ATCC 20344 grown under high and low fumarate producing conditions. The transcriptome was used to construct a database of the ATCC 20344 *in silico* proteome, and the relative protein abundances were obtained by mapping the peptides from the snapshot proteomes against both the ATCC 20344 and RA 99-880 *in silico* proteomes, the latter to account for possible errors in the *de novo* transcript assembly. The experimental setup is outlined in Fig. 3.

The RNA seq reads obtained from the aerobic and anaerobic conditions were combined into one dataset, and assembled *de novo*, resulting in 13'531 contigs (File S2). We used

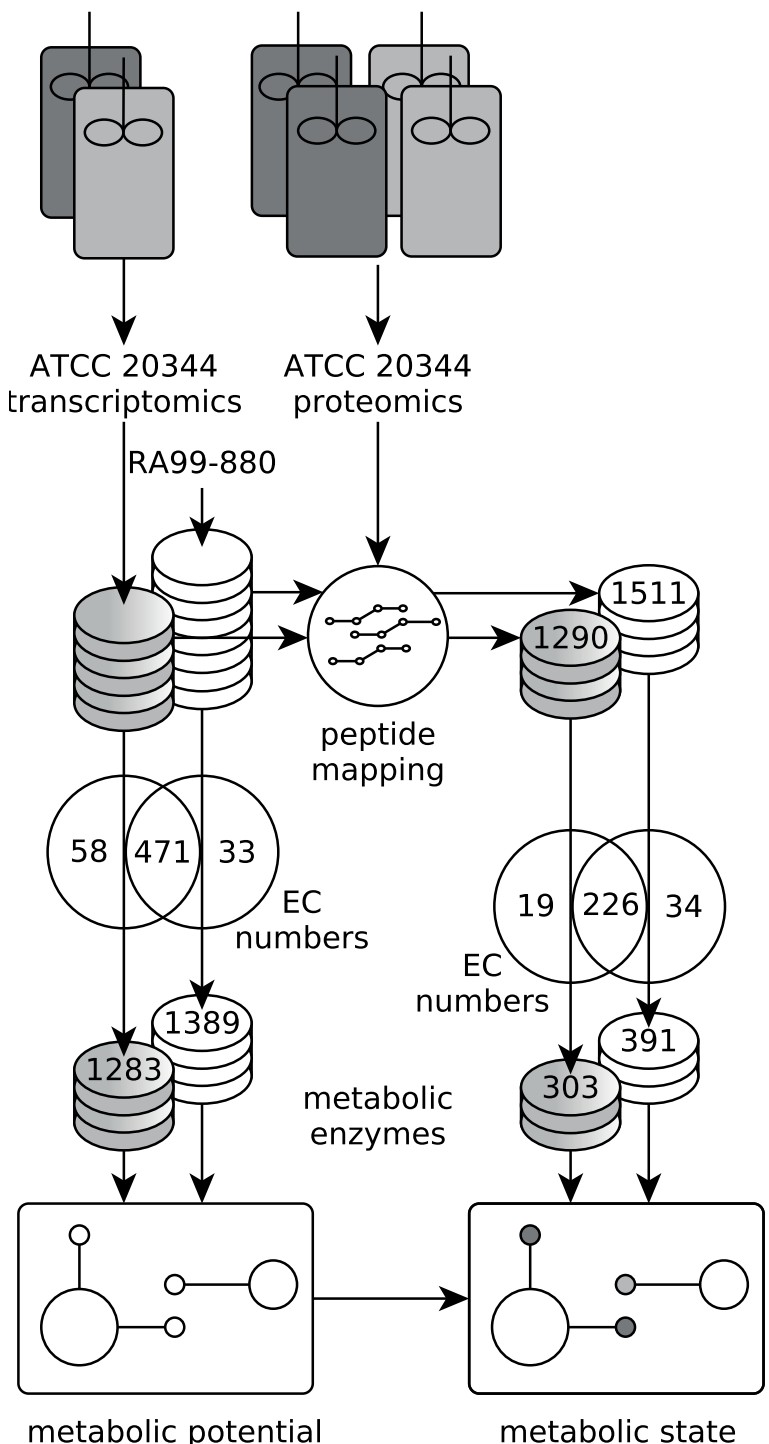

**Figure 3 Experimental setup.** Workflow to establish the metabolic potential and metabolic state of ATCC 20344 grown under high and low fumarate producing conditions. The metabolic enzymes predicted in the ATCC 20344 and RA 99-880 *in silico* proteomes provide a map of the metabolic potential, while the metabolic enzymes identified in the proteomics experiment were used to determine the metabolic state of ATCC 20344 under high and low fumarate producing conditions.

**Table 2** List of enzymes involved in fumarate metabolism with their respective protein abundances under high and low fumarate producing conditions.

| EC number | Consensus protein identifier[a] | Relative protein abundance ± sd [%] | | Log$_2$FC aerobic/anaerobic | Enzyme name |
|---|---|---|---|---|---|
| | | Aerobic | Anaerobic | | |
| 4.2.1.2 | Rd_01690 | 1.51 ± 0.16 | 0.66 ± 0.06 | 1.21 | Fumarate hydratase (fumarase, FumR) |
| 4.3.2.1 | Rd_00962 | 0.09 ± 0.01 | 0.06 ± 0.02 | 0.60 | Argininosuccinate lyase (ASL) |
| 1.3.98.1 | Rd_00873 | 0.04 ± 4e−3 | 0.01 ± 2e−3 | 1.34 | Dihydroorotate dehydrogenase |
| 3.7.1.2 | Rd_01207 | 0.03 ± 3e−3 | 0.00 | – | Fumarylacetoacetase |
| 1.3.5.1 | Rd_01783 | 0.01 ± 2e−3 | 0.00 | – | Succinate dehydrogenase |
| 4.3.2.2 | Rd_00964 | 5e−3 ± 7e−4 | 0.01 ± 8e−7 | −1.59 | Adenylosuccinate lyase |

**Notes.**

[a]Identifiers refer to IDs in Table S4. Note that, where possible, ATCC 20344 enzymes were prioritised. In this case, all enzymes were identified in the ATCC 20344 proteome.

PRIAM (*Claudel-Renard et al., 2003*) to assign EC numbers to the six-frame translation products of the *de novo* contigs, as well as to the RA 99-880 reference proteome (File S3). In ATCC 20344, we predicted 1283 metabolic enzymes, covering 529 EC numbers. In RA 99-880, we predicted 1389 metabolic enzymes, covering 504 EC numbers. The metabolic enzymes were mapped to KEGG pathway maps (*Kanehisa & Goto, 2000*; *Kanehisa et al., 2015*) in order to obtain a rough estimate of the metabolic potential of *R. delemar*.

The proteins obtained from the aerobic and anaerobic conditions were subjected to a shotgun proteomics analysis. A total of 1,290 and 1,511 proteins were identified in the ATCC 20344 and RA 99-880 proteomes, respectively. Roughly one third of the identified proteins comprised metabolic enzymes. A list of all EC numbers predicted in the ATCC 20344 and RA 99-880 *in silico* proteomes, as well as the relative protein abundances and average nucleotide coverages of the proteins and transcripts identified in the experimental conditions (resulting in a total of 1970 proteins, although some are duplicates due to combining ATCC 20344 and RA 99-880 in one file), can be found in Table S4 . The protein pathway coverage (number of ECs per pathway covered by proteins with the predicted function in ATCC 20344 and RA 99-880) is given in Table S5.

## Metabolic pathway enrichment analysis

To determine which pathways play an important role in fumarate accumulation in ATCC 20344, we obtained a list of enzymatic proteins that directly consume or produce fumarate according to the KEGG database (*Kanehisa & Goto, 2000*; *Kanehisa et al., 2015*), and analysed their presence and abundance in the ATCC 20344 high and low fumarate producing conditions (Table 2). In addition, we performed pathway enrichment analysis of differentially expressed enzymes (Table 3 and Table S5).

We found that, amongst the enzymes which interact directly with fumarate and were identified in the ATCC 20344 snapshot proteomes, FumR (EC 4.2.1.2), is the most highly abundant enzyme, both under high and low fumarate producing conditions (Table 2). Although there is no clear enrichment of the TCA cycle enzymes among the differentially expressed metabolic proteins (Table 3), the three enzymes of the reductive TCA cycle, PYC (EC 6.4.1.1, relative protein abundance [% a.u.]: aerobic = 2.14 ± 0.31, anaerobic =

**Table 3** Metabolic pathway enrichment analysis.

| Pathway | # ECs in reference pathway | # ECs (proteins) identified | Differentially expressed | | Overexpressed (aerobic) | | | |
|---|---|---|---|---|---|---|---|---|
| | | | # proteins (1.5-fold) | p-value | # proteins (1.5-fold) | p-value | # proteins (2-fold) | p-value |
| Alanine, aspartate and glutamate metabolism | 50 | 12 (16) | 15 | 0.040 | 10 | 0.102 | 7 | 0.264 |
| Arginine biosynthesis | 32 | 12 (14) | 13 | 0.070 | 10 | 0.033 | 5 | 0.534 |
| beta-Alanine metabolism | 37 | 8 (11) | 9 | 0.389 | 9 | 0.011 | 9 | 0.001 |
| Citrate cycle (TCA cycle) | 25 | 15 (19) | 14 | 0.589 | 10 | 0.292 | 9 | 0.143 |
| Glycolysis/Gluconeogenesis | 49 | 19 (31) | 19 | 0.957 | 10 | 0.946 | 9 | 0.777 |
| Oxidative phosphorylation | 11 | 8 (20) | 16 | 0.327 | 14 | 0.014 | 9 | 0.189 |
| Pyrimidine metabolism | 65 | 11 (13) | 11 | 0.267 | 9 | 0.056 | 6 | 0.243 |
| Pyruvate metabolism | 68 | 15 (23) | 17 | 0.565 | 11 | 0.433 | 11 | 0.102 |
| Valine, leucine and isoleucine degradation | 38 | 10 (14) | 11 | 0.446 | 8 | 0.230 | 8 | 0.055 |

$0.40 \pm 0.02$, $\log_2 FC(\text{aerobic/anaerobic}) = 2.44$), MDH (3 isozymes with EC 1.1.1.37, relative protein abundance [% a.u.]: aerobic $= 0.48 \pm 0.06; 0.15 \pm 0.06; 2.43 \pm 0.08$, anaerobic $= 0.10 \pm 0.01; 0.02 \pm 0.00; 1.85 \pm 0.09$, $\log_2 FC(\text{aerobic/anaerobic}) = 2.20; 3.19; 0.40$), and FumR (EC 4.2.1.2, see Table 2 for relative protein abundance), are all overexpressed in the high fumarate producing condition.

The second most highly abundant protein related to fumarate metabolism is argininosuccinate lyase (ASL) (EC 4.3.2.1). ASL is a urea cycle enzyme involved in arginine biosynthesis. The arginine biosynthesis pathway showed a significant number of differentially expressed proteins (Table 3), and we found that the enzymes comprising the urea cycle are overexpressed in the high fumarate producing condition (Table 4). This suggests that in ATCC 20344, the urea cycle plays an important role in fumarate accumulation. Most interestingly, ASL and FUM constitute a crucial link between carbon and nitrogen metabolism by connecting the TCA- and urea cycles (also referred to as "Krebs bicycle").

The observed protein abundances for FumR and ASL offer an explanation for the importance of a high carbon:nitrogen ratio for fumarate accumulation in *Rhizopus spp* (*Magnuson & Lasure, 2004*; *Goldberg, Rokem & Pines, 2006*). In humans, starvation induces a net breakdown of stored energy sources, starting with fatty acids and, when exposed to prolonged starving conditions, proteins from muscle tissue (*Felig, 1973*). The degradation of protein, or amino acids, results in the liberation of ammonia, which is then carried to the urea cycle as L-glutamate (*Feillet & Leonard, 1998*). In the urea cycle, the L-glutamate is converted to urea, which is subsequently excreted (*Feillet & Leonard, 1998*). Based on the significant enrichment scores of pathways involved in amino acid metabolism (Table 3), we propose that the nitrogen starvation, induced by the transfer of ATCC 20344 from growth- to production medium, triggers a similar switch in metabolism, and amino acid catabolism starts to occur. The resulting fluxes through the urea cycle yield an excess of fumarate (Fig. 4).

**Table 4** Urea cycle enzymes with their respective protein abundances under high and low fumarate producing conditions.

| EC number | Consensus protein identifier[a] (ref) | Relative protein abundance ± sd [%] | | Log$_2$FC aerobic/anaerobic | Enzyme name |
|---|---|---|---|---|---|
| | | aerobic | anaerobic | | |
| 2.1.3.3 | Rd_01058 (A) | 0.03 ± 0.01 | 0.02 ± 8e−3 | 0.51 | Ornithine carbamoyltransferase (OTC) |
| 6.3.4.5 | Rd_01708 (R) | 0.86 ± 0.07 | 0.57 ± 0.05 | 0.59 | Argininosuccinate synthase (ASS) |
| 6.3.4.5 | Rd_01709 (R) | 0.00 | 0.04 ± 0.02 | – | Argininosuccinate synthase (ASS) |
| 4.3.2.1 | Rd_00962 (A) | 0.09 ± 0.01 | 0.06 ± 0.02 | 0.60 | Argininosuccinate lyase (ASL) |
| 3.5.3.1 | Rd_00988 (R) | 0.13 ± 0.02 | 0.07 ± 0.01 | 0.97 | Arginase (ARG) |
| 3.5.3.1 | Rd_00989 (R) | 0.19 ± 0.02 | 0.11 ± 0.03 | 0.76 | Arginase (ARG) |

**Notes.**

[a]Identifiers refer to IDs in Table S4. Note that, where possible, ATCC 20344 enzymes were prioritised. Letters in brackets refer to the reference proteome, with A = ATCC 20344 and R = RA 99-880.

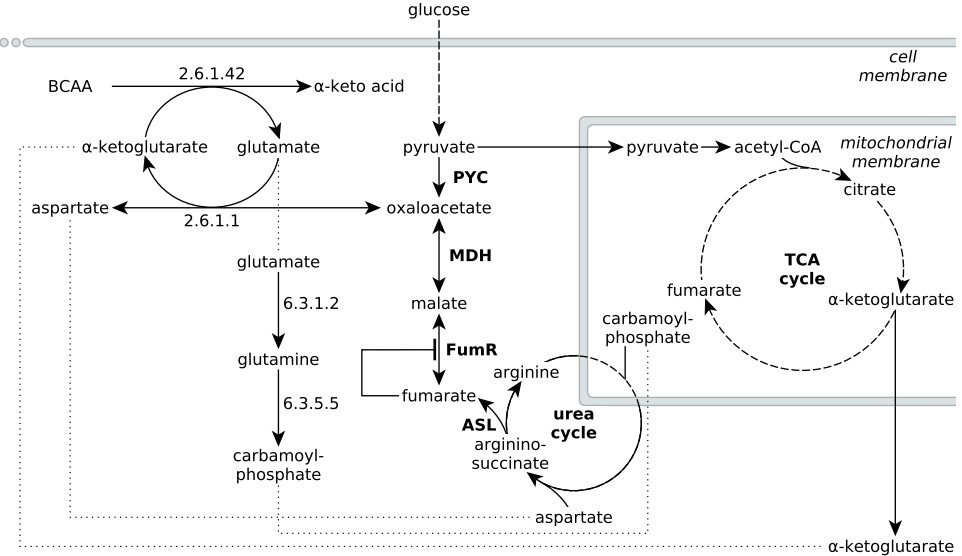

**Figure 4** **Extended network of metabolic pathways involved in fumarate metabolism in *R. delemar*.** The extended model of fumarate accumulation reconstructed from the ATCC 20344 snapshot proteomes under high and low fumarate producing conditions takes the formation of fumarate *via* the urea cycle into account.

Specifically, the pathway for the degradation of the branched-chain amino acids (BCAA) valine, leucine and isoleucine shows a significant number of overexpressed enzymes in the high fumarate producing condition. BCAA catabolism is initiated by BCAA aminotransferase (EC 2.6.1.42), which catalyses the transfer of an amino group from any of the three BCAAs to $\alpha$-ketoglutarate, yielding L-glutamate and the respective $\alpha$-keto acid as products (Fig. 4). Under starvation conditions in which both the nitrogen and carbon source in the culture medium are limited, the carbon skeletons of the deaminated amino acids can be used to replenish acetyl-CoA (from leucine), or the TCA cycle intermediate succinyl-CoA (from valine) or both of these metabolites (from isoleucine), and thereby, ultimately, to generate energy for growth. Under conditions of excess carbon and limited

nitrogen, however, it is unlikely that amino acid catabolism is driven by energy demand of the organism. More importantly, it is crucial for *R. delemar* grown under nitrogen depleted conditions to decouple carbon catabolism from cell proliferation, as there is little nitrogen available for *de novo* protein biosynthesis.

One way to decouple carbon catabolism from biomass formation is by reducing the amount of ATP generated. Under aerobic growth conditions, ATP is generated *via* oxidative phosphorylation. Under anaerobic conditions, *R. delemar* generates ATP *via* ethanol fermentation (Fig. 1). Another option for ATP generation is alternative respiration, mediated by the key enzyme alternative oxidase (AOX). AOX diverts the electrons passing through the electron transport chain in the mitochondria at the ubiquinone pool and transfers them directly to oxygen, thereby bypassing the oxidative phosphorylation complexes III and IV, resulting in an overall lower ATP yield. *Gu et al. (2014)* found that the activity of AOX is positively correlated with fumarate production in *R. delemar*. In contrast, we identified AOX (Rd_00967 (A)) only in the snapshot proteome of the anaerobic condition. The transcriptomics measurements further underpin our proteomics results, since *aox* was overexpressed (>4-fold) in the anaerobic condition (Table S4). This might seem counterintuitive at first, since the expression of *aox* is generally regarded as a means of dealing with increased oxidative stress. However, the electron flow through AOX has been found to be inversely proportional to nitrogen availability in various different plant systems (*Vanlerberghe, 2013*). The increase of AOX under nitrogen limited conditions and the resulting decrease of the respiratory ATP yield have been associated with a deliberately reduced efficiency in converting carbon to biomass; by using the non-energy conserving AOX, the "redundant" carbohydrate can be metabolised without being coupled to growth.

Another way of decoupling carbon catabolism from energy generation, and thereby cell proliferation, is channelling the products from amino acid degradation to the mitochondria for mitochondrial protein synthesis, and thereby away from cytosolic protein synthesis; a mechanism suggested to take place under nitrogen starvation conditions in human cells, irrespective of the D-glucose availability (*Johnson et al., 2014*). In this, amino acid catabolism is the first step to adapt to nitrogen limiting conditions, and we propose that, in a similar mechanism, the urea cycle plays a key role for the accumulation of fumarate in *R. delemar*. This is supported by the work of *Chen et al. (2015)* who showed that, from a range of selected enzymes, overexpressing ASL, while keeping the expression of adenylosuccinate lyase low, resulted in the highest fumarate titer in *Torulopsis glabrata*.

## CONCLUSIONS

The accumulation of fumarate in the natural fumarate producer *R. delemar* has been mostly attributed to the consecutive conversion of pyruvate to oxaloacetate, L-malate and fumarate by cytosolic enzymes of the reductive TCA cycle. In addition, our proteomics data have revealed that the nitrogen-limitation under fumarate producing conditions induces amino acid catabolism, which leads to an increased flux through the urea cycle. Further investigation is required to verify the involvement of the urea cycle in *R. delemar* fumarate accumulation. As *R. delemar* can utilise urea as nitrogen source, higher fluxes through the urea cycle will not necessarily lead to a measurable increase of urea in the medium. Our

comparative proteomics analysis of high and low fumarate producing conditions in *R. delemar* ATCC 20344 has resulted in a novel holistic view on fumarate production that expands the knowledge on fumarate production in this fungus, and provides a basis for further biochemical explorations regarding biotechnological fumarate production.

## ACKNOWLEDGEMENTS

We would like to thank Sybe Hartmans, Michael Volpers and Brendan Ryback for critically reading and commenting on draft versions of the manuscript. In addition, we would like to thank Bastian Hornung for valuable input for the RNA seq data analysis, Ruud Weusthuis for valuable feedback on general aspects of this manuscript, and Tom Schonewille and Merlijn van Gaal for their contribution in the experimental part of this work.

### Funding

This work has been carried out on a basis of a grant in the framework of the WUR IPOP Systems Biology program KB-17-003.02-026 "Genome-wide metabolic modelling and data integration of organic acid production in filamentous fungi" (DIO) and the BE-Basic (http://www.be-basic.org/) program F01.002 "Itaconic/fumaric acids: Novel economic and eco-efficient processes for the production of itaconic and fumaric acid" (JATR). The funders had no role in study design, data collection and analysis, decision to publish, or preparation of the manuscript.

### Grant Disclosures

The following grant information was disclosed by the authors:
WUR IPOP Systems Biology program: KB-17-003.02-026.

### Competing Interests

The authors declare there are no competing interests.

### Author Contributions

- Dorett I. Odoni conceived and designed the experiments, analyzed the data, wrote the paper, prepared figures and/or tables, reviewed drafts of the paper.
- Juan A. Tamayo-Ramos conceived and designed the experiments, performed the experiments, analyzed the data, wrote the paper, reviewed drafts of the paper.
- Jasper Sloothaak performed the experiments, analyzed the data, reviewed drafts of the paper.
- Ruben G.A van Heck analyzed the data, reviewed drafts of the paper.
- Vitor A.P Martins dos Santos conceived and designed the experiments, contributed reagents/materials/analysis tools, reviewed drafts of the paper.
- Leo H. de Graaff conceived and designed the experiments, contributed reagents/materials/analysis tools, wrote the paper, reviewed drafts of the paper.
- Maria Suarez-Diez and Peter J. Schaap conceived and designed the experiments, reviewed drafts of the paper.

## Data Availability

Proteomics data:

PRIDE: http://www.ebi.ac.uk/pride/archive/projects/PXD004600

RNA seq data: EMBL-EBI, PRJEB14210, http://www.ebi.ac.uk/ena/data/view/PRJEB14210.

## Supplemental Information

Supplemental information for this article can be found online at http://dx.doi.org/10.7717/peerj.3133#supplemental-information.

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
