# Peer review of "Comparative proteomics of Rhizopus delemar ATCC 20344 unravels the role of amino acid catabolism in fumarate accumulation"

_PeerJ, doi:10.7717/peerj.3133_

## Round 0.1 · original submission · Minor Revisions

Please address all issues indicated by both reviewers.

Reviewer 1 ·

Basic reporting

In this manuscript, the authors studied metabolic pathways involved in fumarate production in Rhizopus delemar ATCC 20344. Transcriptomics and proteomics approaches were applied to reveal relevant components in fumarate accumulation by comparing nitrogen starved ATCC 20344 under aerobic and anaerobic conditions. Data confirms the presence of directly linked proteins in fumarate production. Data also indicates other amino acid catabolism pathways may interplay in fumarate accumulation. The comparative proteomics data is quite informative, which provides deeper understanding to mechanism of fumarate biosynthesis in ATCC 20344, but also opens a gate to researchers who are interested in any metabolic pathways in Rhizopus delemar.

Experimental design

The study is well planned and well carried out.

Validity of the findings

The conclusion is largely based on the screen data, which needs further validation with biochemical approaches.

Additional comments

Specific concerns:
1. The authors proposed “The resulting fluxes through the urea cycle yield an excess of fumarate.” It is based on the screen data from proteomics study. As urea cycle stands out and it shows in the last sentence of abstract, it needs more support. Have you confirmed the involvement of urea cycle biochemically, like measuring the content of urea in aerobic/anaerobic conditions?
2. Line 328-334: These proteins are discussed in Fig. 1 and readers would be curious about how they act quantitatively in two studied conditions. I have found PYC (2.14±0.31/0.40±0.02/2.44) is quite interesting only after taking an effort to locate it in a large supplementary excel file. I suggest that you have those presented and discussed in main text.
3. Fig2, Line 242: “allowed ATCC 20344 to grow at a normal rate,” How to define “a normal rate”? Is it a same rate as under aerobic condition? Did you count the cell population before and after (over 100 hours) the fermentation? Were cells still alive? I would like to see fumarate yield in “g/g ATCC 20344” if possible.
4. The supplementary data in excel files is carefully organized. A title will make it better.
5. Line 351-357, Line370-382: Please provide references.

Thank you!

·

Basic reporting

* Based on the information provided below, I checked the availability of the deposited raw data as required by PeerJ but failed to login. I just tried three times on February 2 of 2017.

"
Your raw data must be accessible at this location for review.
Proteomics data:
1) PRIDE
2) PXD004600
3) http://www.ebi.ac.uk/pride/archive/login
Accession details for review purposes:
Username: reviewer14653@ebi.ac.uk
Password: r1CpiwSv
"

Experimental design

* Line148: The authors mentioned that De novo assembly of the reads was performed using the IDBA-UD assembler v1.1. Although IDBA-UD demonstraed good performance in some benchmarks (e.g. http://cab.spbu.ru/software/spades/#benchmark), there are three separate assemblers in IDBA-UD package, and IDBA-Tran should be used for transcriptome data (https://github.com/loneknightpy/idba). Please clearly described the name and version of the tools used in the work if the authors actually used IDBA-Tran in the work.

* The different combinations of parameters used with the same sequence assembler on the same data usually produce quite different assemly results. I suggest the authors describe in detail (name, version number and parameters in a supplementary file) the software tools and the data used in each step in their RNA-seq sequence assembling and the downstream analysis, which could give the authors and other researchers a chance in the future to reproduce their results based on the available raw data.

* Line217 and 301: In order that the future readers could reproduce the results, the authors should prepare a detailed method description (in a supplementary file) to describle how they assigned (mapped) enzymes to KEGG pathways.

Validity of the findings

* Line297: The authors should explain more clear how EC numbers are assigned to the predicted proteins of the de novo contigs and to the RA 99-880 reference proteome. Namely, how the authors predicted (assigned) the protein function (EC number) of the predicted (six-frame translated) proteome and RA 99-880 reference proteome.

* Line310: The collumn C in table S3 shows the number of proteins is 1970. I suggest the authors to briefly explain the relationship between this number and the numbers (1283, 1389, 1290, 1511) of proteins identified in Figure 3.

* Line326: Table 3 demonstrated the significant increase of expression of the identified enzymes in the relevant metabolic pathways. The 2nd collumn (# ECs in reference pathway) and the 3rd collumn (# proteins (ECs) identified) could be more clear if the authors exchange the # ECs in brackets and the # proteins outside the brackets (for example, changing 16(12) into 12(16) as both 50 in 2nd collumn and 12 in 3rd collumn are # EC).

Additional comments

Although the manuscript needs some improvments in the details of method, data and results, but the authors presented the interesting findings in the economically important Rhizopus delemar ATCC 20344 and made a reasonable hypothesis with the data support (proteomic data accompanied by transcriptomic data) that the nitrogen starvation might induce an alternative amino acid metabolic pathway and then the pathway may contribute the fumarate accumulation. Although PeerJ does not require the novelty of the submitted research work but the authors presented a novel work and the community will benefit from their work.

---

## Round 0.2 · accepted · Accept

I would like to thank you for your efforts to address reviewers' comments.